# Targeting serum phosphate trajectory stratification to improve outcomes in high-risk Cardiovascular-Kidney-Metabolic-Sepsis cohorts

Jinwei Dai[1,2☯], Wenye Xu[1,2☯], Nianzhe Sun[2,3], Ting Wu[2,4*], Zhaoxin Qian[1,2*], Zhihong Zuo [1,2*]

1 Department of Critical Care Medicine, Xiangya Hospital, Central South University, Changsha, Hunan, China, 2 National Clinical Research Center of Geriatric Disorders, Xiangya Hospital, Central South University, Changsha, China, 3 Department of Orthopedics, Hand & Microsurgery, The Xiangya Hospital of Central South University, Changsha, Hunan, China, 4 Department of cardiovascular medicine, The Xiangya Hospital of Central South University, Changsha, Hunan, China

☯ These authors contributed equally to this work.
* Wuting@csu.edu.cn (TW); xyqzx@csu.edu.cn (ZQ); China.zhihong.zuo1995@gmail.com (ZZ)

## Abstract

### Background

Sepsis patients exhibit complex clinical conditions, frequently complicated with metabolic dysregulation. Cardiovascular-Kidney-Metabolic Syndrome (C-K-M) is classified as below: stage 0, no C-K-M risk factors; stage 1, excess or dysfunctional adiposity; stage 2, metabolic risk factors (hypertriglyceridemia, hypertension, diabetes, metabolic syndrome) or moderate- to high-risk chronic kidney disease; stage 3, subclinical cardiovascular diseases (CVD) in C-K-M syndrome or risk equivalents (high predicted CVD risk or very high-risk chronic kidney diseases); and stage 4, clinical CVD in C-K-M syndrome. While high-risk patients defined by C-K-M criteria often have poor outcomes, studies seldom have classified these patients into subtypes based on metabolic profiles. Serum phosphate, recently recognized as a potential metabolic and organ function marker, has unclear dynamic trajectories and prognostic significance across high-risk CKM-sepsis subgroups.

### Purpose

This study aimed to evaluate the association between serum phosphate trajectories and clinical prognosis, specifically 28-day mortality, among high-risk C-K-M-sepsis patients and across various subgroups.

### Methods

We extracted data for high-risk C-K-M-Sepsis patients from the MIMIC-IV database. After developing a simplified C-K-M staging system, we used unsupervised

**Data availability statement:** The data used in this study were obtained from the Medical Information Mart for Intensive Care IV (MIMIC-IV, version 3.1) database, which is publicly available at https://physionet.org/content/mimiciv/3.1/. Access to the database requires registration for a PhysioNet account, completion of the Collaborative Institutional Training Initiative (CITI) "Data or Specimens Only Research" course, and approval of a data use request.

**Funding:** This work was supported by Funding was provided by the National Key Research and Development Program of China (2022YFC2009800), the Natural Science Foundation of Changsha(kq2403030), the Natural Science Foundation of Hunan Province(2024JJ6659). There was no additional external funding received for this study.

**Competing interests:** The authors declare that the research was conducted in the absence of any commercial or financial relationships that could be construed as a potential conflict of interest.

consensus clustering to identify four metabolic phenotypes. Serum phosphate trajectories during the first seven ICU days were summarized by daily earliest measurements. Associations between phosphate trajectory clusters and 28-day ICU mortality were examined using multivariable logistic regression, inverse probability weighting (IPW) derived from propensity scores, and doubly robust estimation. Subgroup analyses stratified by age, sex, and key comorbidities were conducted, and results were visualized as forest plots.

## Results

Multivariate analysis revealed that trajectory Group 3 (persistently high serum phosphate) had significantly increased mortality risk (OR=2.909, 95% CI: 2.121–2.991, p<0.001). Elevated risk was prominent in younger (<65 years) and male subgroups. Comorbidity analysis identified CVA and COPD as significant risk factors.

## Conclusion

Serum phosphate trajectory patterns significantly correlate with 28-day mortality in high-risk CKM-sepsis patients, highlighting potential distinct metabolic phenotypes. Early intervention targeting serum phosphate levels may improve prognosis in high-risk subgroups.

## Introduction

Sepsis is a major cause of high mortality worldwide, and its pathophysiology involves complex immune dysregulation, metabolic disturbances, and organ dysfunction [1,2]. Cardiovascular-Kidney-Metabolic Syndrome" (C-K-M Syndrome), a novel concept introduced by the American Heart Association in 2023, incorporates an all-organ-system approach into disease management. C-K-M syndrome is defined as a health disorder attributable to connections among obesity, diabetes, chronic kidney disease (CKD), and cardiovascular disease (CVD), including heart failure, atrial fibrillation, coronary heart disease, stroke, and peripheral artery disease [3]. This innovative paradigm has attracted widespread attention. Multiorgan dysfunction and metabolic disturbances are key characteristics of sepsis [4], and the incorporation of the C-K-M criteria into managing the disease course of sepsis is of significant importance.

In recent years, serum phosphate levels, as a routinely measured biomarker, have gradually garnered clinical attention [5]. Hyperphosphatemia has been described in sepsis, metabolic or respiratory alkalosis and refeeding syndrome [6]. Research has shown that serum phosphate plays a critical role in energy metabolism, cellular signal transduction, and acid-base balance. Its abnormal fluctuations may reflect metabolic disturbances and organ dysfunction [7–9]. Previously, higher serum phosphate levels were indicated to be associated with adverse outcomes in various diseases, including CKD, CVD and blunt trauma [10–12]. Among patients with sepsis, serum phosphate disturbances contribute to worse outcomes [13].

Furthermore, serum phosphate levels were positively and independently associated with 28-day mortality in septic shock [14]. However, studies on the dynamic patterns of serum phosphate in sepsis patients and its relationship with clinical outcomes remain insufficient.

Recent studies have elucidated that dysregulated phosphate metabolism may impact cardiovascular-kidney-metabolic (CKM) syndrome via multiple pathophysiological mechanisms. Hyperphosphatemia can induce endothelial dysfunction and vascular calcification, thus exacerbating cardiovascular risks. Moreover, abnormal serum phosphate levels could interfere with insulin signaling pathways, intensifying metabolic disorders [15]. Additionally, phosphate dysregulation is associated with elevated levels of fibroblast growth factor-23 (FGF23), closely linked to the progression of chronic kidney disease and cardiovascular disease [16]. Hence, a deeper understanding of phosphate's role in CKM syndrome is vital for assessing and managing high-risk patients effectively.

Based on the MIMIC-IV database, to explore the relationship of dynamic changes of serum phosphate and sepsis, this study firstly screened CKM-Sepsis comorbid patients and ultilized multiple statistical methods to identify High-Risk CKM-Sepsis patients and different subtypes, providing a foundation for early risk stratification and personalized treatment.

## Materials and methods

### Data extraction and patient selection

The data used in this study were obtained from MIMIC-IV (3.1) (https://mimic.mit.edu), a large database that records clinical information of patients. The database includes information on patients admitted to the intensive care unit (ICU) at Beth Israel Deaconess Medical Center (BIDMC) in Boston, Massachusetts, USA. The BIDMC Institutional Review Board approved a waiver of informed consent and the sharing of research resources. The author (JW. D) obtained access to the database (certification number: 62317039).

Inclusion Criteria: Patients aged ≥ 18 years; Patients meeting Sepsis 3.0 criteria.

Exclusion Criteria: Patients with an ICU stay of less than 24 hours.; Patients lacking records for triglycerides, creatinine, or serum phosphate; For patients with multiple ICU admissions, only data from the first hospitalization were included.

Baseline patient characteristics were obtained using Structured Query Language (SQL) and PostgreSQL (version 14.2). These attributes include demographic details such as age, gender, body mass index (BMI), and race. In addition, vital signs such as heart rate (HR), systolic blood pressure (SBP), diastolic blood pressure (DBP), mean arterial pressure (MAP), arterial oxygen saturation (SpO2), and temperature (T) were recorded. The severity of illness at admission was assessed using the Sequential Organ Failure Assessment (SOFA) score, Acute Physiology Score III (APS III), Systemic Inflammatory Response Syndrome (SIRS) score, Simplified Acute Physiology Score II (SAPSII), Oxford Acute Severity of Illness Score (OASIS), and Glasgow Coma Scale (GCS). Laboratory test results included red blood cell (RBC) count, white blood cell (WBC) count, platelet count, hemoglobin level, albumin concentration, serum creatinine (Scr) level, as well as sodium, potassium, and calcium ion concentrations. In addition, fasting blood glucose (FBG) levels, glycated hemoglobin (HbA1c), anion gap, serum phosphate, lactate, triglycerides (TG), total cholesterol (TC), high-density lipo-protein cholesterol (HDL-C), low-density lipoprotein cholesterol (LDL-C), alanine aminotransferase (ALT), and aspartate aminotransferase (AST) were obtained. Information regarding the use of antidiabetic, antihypertensive, and lipid-lowering medications was also collected. Furthermore, the following comorbidities were extracted from the MIMIC-IV database: coronary heart disease (CHD), congestive heart failure (CHF), myocardial infarction (MI), hypertension, diabetes, hyper-lipidemia, cerebrovascular accident (CVA), peripheral arterial disease (PAD), atrial fibrillation (AF), chronic kidney disease (CKD), acute kidney injury (AKI), chronic obstructive pulmonary disease (COPD), respiratory failure (RF), stroke, liver disease (LD), pneumonia, sepsis, and cancer. The primary outcome of this study was the incidence of 28-day ICU mortality. Secondary outcomes included ICU length of stay.

The CKD-EPI equation for estimating GFR, developed in 2021 (in ml/min/1.73 m²), does not incorporate a race coefficient [1]:

$$eGFR = \begin{cases} 143 \times \left(\frac{SCr}{0.7}\right)^{-0.241} \times 0.993^{age}, & \text{if female and } SCr \leq 0.7 \text{ mg/dl}, \\ 143 \times \left(\frac{SCr}{0.7}\right)^{-1.200} \times 0.993^{age}, & \text{if female and } SCr > 0.7 \text{ mg/dl}, \\ 142 \times \left(\frac{SCr}{0.9}\right)^{-0.302} \times 0.993^{age}, & \text{if male and } SCr \leq 0.9 \text{ mg/dl}, \\ 142 \times \left(\frac{SCr}{0.9}\right)^{-1.200} \times 0.993^{age}, & \text{if male and } SCr > 0.9 \text{ mg/dl}. \end{cases} \tag{1.1}$$

To stratify the High-Risk CKM-Sepsis cohort, we developed a simplified CKM staging system based on baseline clinical and laboratory parameters. This system was designed to capture the severity of metabolic derangement and organ dysfunction and comprised the following stages: Stage 0: Patients with normal BMI (18.5 ≤ BMI < 25), normal blood glucose (glucose < 100 mg/dL), (SBP < 120 mmHg and NBP < 80 mmHg), and triglycerides < 135 mg/dL, with no evidence of CKD, HF, MI, CVA, PAD, or atrial AF. Stage 1: Patients with BMI ≥ 25 or those with mildly elevated blood glucose (between 100 and 124 mg/dL) or receiving antidiabetic therapy. Stage 2: Patients exhibiting one or more of the following: triglycerides ≥ 135 mg/dL or receiving lipid-lowering therapy; elevated blood pressure (SBP ≥ 130 mmHg or NBP ≥ 80 mmHg) or receiving antihypertensive therapy; diagnosis of type 2 diabetes mellitus (T2DM) or glucose ≥ 100 mg/dL or receiving antidiabetic therapy; or reduced kidney function (eGFR < 60) or presence of CKD. Stage 3: Patients with HF in the absence of other major cardiac events, or with eGFR ≤ 30 mL/min/1.73 m². Stage 4: Patients with any of the following cardiac conditions: HF, MI, CVA, PAD, or AF, in combination with one or more metabolic risk factors (elevated triglycerides, hypertension, diabetes, reduced eGFR, or CKD). Stage 4 was further subdivided into: Stage 4a: eGFR > 15 mL/min/1.73 m². Stage 4b: eGFR ≤ 15 mL/min/1.73 m².

## Statistical analysis

All analyses were performed in R (version 4.1.2). Categorical variables were reported as counts and percentages, and between-group comparisons were carried out using the chi-square test or Fisher's exact test, as appropriate. Continuous variables were expressed as mean ± standard deviation or median (interquartile range) and compared across groups by one-way analysis of variance when normally distributed or by the Kruskal–Wallis test otherwise.

## Simplified CKM staging system

We constructed a simplified CKM staging algorithm to further stratify high-risk CKM-Sepsis patients. This staging incorporated baseline body mass index, fasting glucose, blood pressure, triglycerides, and the presence of comorbidities (CKD, heart failure, myocardial infarction, cerebrovascular accident, peripheral arterial disease, and atrial fibrillation). Patients were assigned to Stage 0 (4, with Stage 4 subdivided into 4a and 4b based on eGFR thresholds.

## Unsupervised consensus clustering

Machine learning (ML) classifier models have become important tools for identifying disease subtypes [17]. To determine the clinical phenotypes of patients with High-Risk Cardiovascular-Kidney-Metabolic-Sepsis, we applied unsupervised ML methods for consensus clustering to identify clinical phenotypes. This method performs clustering analysis by reducing the data dimension, and k-means is the most commonly used type among them [18]. We used a pre-specified 80% subsampling parameter and 100 iterations, and assigned the number of potential clusters (k) to a range from 2 to 10. The optimal number of clusters was determined by examining the clustering consistency plots in the consensus matrix (CM) heatmap, cumulative distribution function (CDF), within-cluster consistency scores, and the proportion of pairs of ambiguous clusters (PAC). The final number of clusters was determined to be 4. Ultimately, four subtypes were identified.

## Serum phosphate trajectory modeling

Serum phosphate levels within the first 7 ICU days were consolidated by selecting the earliest daily measurement for each patient. Within the high-risk cohort, we performed group-based trajectory modeling on the 7-day serum phosphate series

to identify distinct longitudinal patterns. We next assessed the independent relationship between serum phosphate trajectory patterns and 28-day ICU mortality using a comprehensive multivariable and propensity score-based framework. First, we fitted a multivariable logistic regression model (with the "Low-Stable" trajectory as the reference category) adjusting for age, sex, body mass index, SOFA score, chronic kidney disease, heart failure, myocardial infarction, cerebrovascular accident, peripheral arterial disease, atrial fibrillation, type 2 and type 1 diabetes, chronic obstructive pulmonary disease, first-day lactate, triglycerides, estimated glomerular filtration rate, and glucose. Second, we derived propensity scores for membership in each phosphate trajectory cluster via logistic regression on the same covariates and applied inverse probability weighting (IPW) to create a pseudo-population in which baseline characteristics were balanced; these stabilized weights were then used in a weighted logistic model. Third, we implemented a doubly robust approach that combines the outcome model and IPW to further guard against model misspecification. Finally, to explore effect modification, we conducted subgroup analyses stratified by age (< 65 vs. ≥ 65), sex, and key comorbidities. Adjusted odds ratios, 95% confidence intervals, and p-values from all models were visualized in forest plots generated with ggplot2.

Statistical significance was defined as a two-sided p-value < 0.05. All baseline characteristic tables were generated using the TableOne package and further refined with kableExtra to produce publication-quality tables.

## Results

### Patient Cohort

S2 Graphic abstract provides a brief overview of this study. After a rigorous screening process, a total of 4,929 sepsis patients were included (Fig 1). Simultaneously, a simplified CKM staging system was developed using baseline clinical and laboratory data, incorporating criteria based on BMI, blood glucose, blood pressure, triglyceride levels, and the presence of comorbidities. Patients were stratified into Stage 0 through Stage 4 (with Stage 4 further subdivided into 4a and

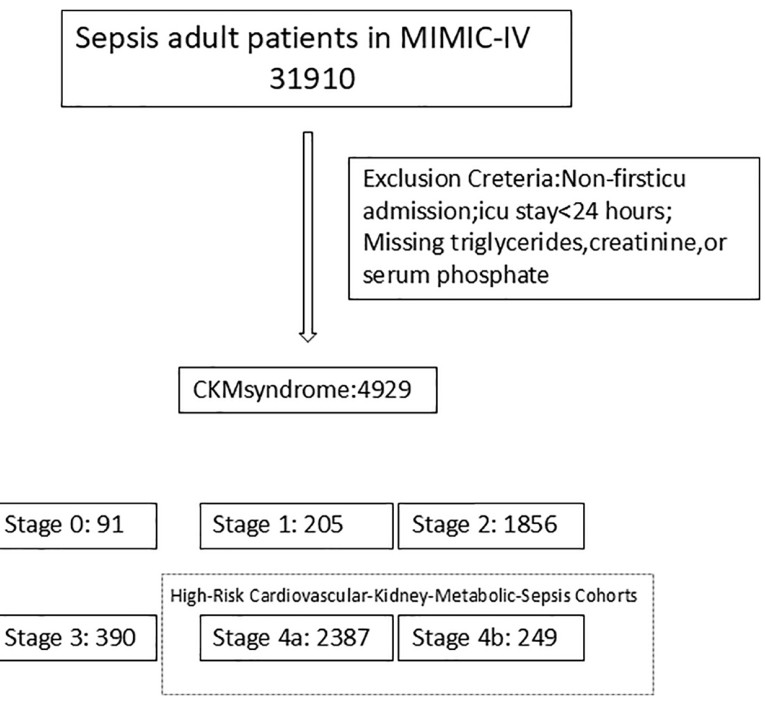

**Fig 1. Patient Selection Flowchart.** Flowchart of patient selection.

4b based on eGFR). Notably, after manual re-assignment of ambiguous cases, the final distribution of CKM stages was as follows: Stage 0 (n = 91), Stage 1 (n = 205), Stage 2 (n = 1,856), Stage 3 (n = 390), Stage 4a (n = 2,387), and Stage 4b (n = 249). Most sepsis patients fall within Stage 2 to Stage 4b. We defined Stage 0–1 as low risk, Stage 2–3 as moderate risk, and Stage 4a–4b as high risk. Fig 2 illustrates the distribution of patients as well as the 28-day ICU mortality rates across different risk groups. A total of 2,636 High-Risk CKM-Sepsis patients (Stage 4a (n = 2,387), and Stage 4b (n = 249) were identified from the MIMIC database after applying our inclusion and exclusion criteria.

## Consensus clustering and CKM staging

Table 1 presents the baseline characteristics of the four subtypes of High-Risk Sepsis–CKM patients identified through consensus clustering (**S1 Fig**). The study cohort comprised 720 patients in Group 1, 737 in Group 2, 331 in Group 3, and 848 in Group 4. Significant differences were observed across groups for multiple variables (all p < 0.001). Age varied substantially among clusters, with Group 2 patients being the youngest (mean 56.99 ± 11.46 years), whereas Groups 1 and 4 had higher mean ages of 70.92 ± 11.70 and 78.79 ± 8.46 years, respectively. Gender distribution also differed significantly, with a higher proportion of females in Group 4 (47.9%) compared to the other groups. Organ dysfunction scores reflected pronounced disparities: Group 3 demonstrated the highest SOFA (11.35 ± 3.71) and SAPSII (57.59 ± 14.86) scores, in contrast to the lower scores observed in Group 2 (SOFA 4.70 ± 2.65; SAPSII

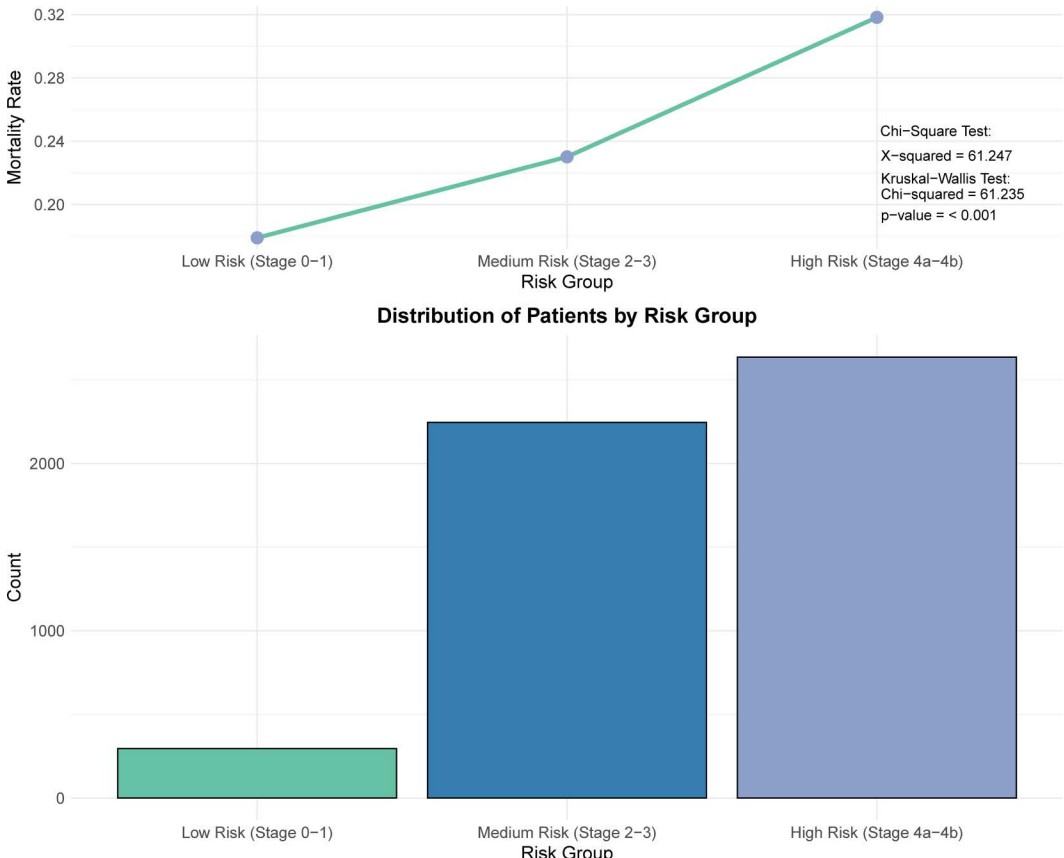

**Fig 2. CKM Staging-based Patient Categorization and Mortality.** Patients are categorized into low-risk (Stage 0–1), medium-risk (Stage 2–3), and high-risk (Stage 4a–4b) groups based on CKM staging. The bar chart shows the number of patients in each group, while the line chart displays corresponding mortality rates. Both the chi-square test and Kruskal–Wallis test indicate statistically significant differences among the groups (p < 0.001).

**Table 1. Baseline Characteristics of High-Risk CKM-Sepsis Patients.**

| Variable | level | 1 | 2 | 3 | 4 | p |
|---|---|---|---|---|---|---|
| n | | 720 | 737 | 331 | 848 | |
| age (mean (SD)) | | 70.92 (11.70) | 56.99 (11.46) | 63.53 (13.21) | 78.79 (8.46) | <0.001 |
| gender | Female | 278 (38.6) | 248 (33.6) | 98 (29.6) | 406 (47.9) | <0.001 |
| | Male | 442 (61.4) | 489 (66.4) | 233 (70.4) | 442 (52.1) | |
| race | WHITE | 390 (54.2) | 362 (49.1) | 155 (46.8) | 517 (61.0) | <0.001 |
| | OTHER | 330 (45.8) | 375 (50.9) | 176 (53.2) | 331 (39.0) | |
| weight (mean (SD)) | | 90.08 (28.59) | 91.96 (30.37) | 95.34 (30.37) | 79.06 (21.45) | <0.001 |
| height (mean (SD)) | | 169.54 (9.16) | 171.01 (9.44) | 171.62 (9.66) | 168.18 (8.90) | <0.001 |
| BMI (mean (SD)) | | 31.29 (9.60) | 31.36 (9.82) | 32.21 (9.39) | 27.90 (7.13) | <0.001 |
| sofa (mean (SD)) | | 9.22 (3.13) | 4.70 (2.65) | 11.35 (3.71) | 4.91 (2.45) | <0.001 |
| sapsii (mean (SD)) | | 53.45 (13.08) | 31.95 (9.30) | 57.59 (14.86) | 41.60 (9.21) | <0.001 |
| lactate (mean (SD)) | | 2.69 (1.77) | 1.93 (1.22) | 4.78 (4.28) | 1.70 (0.79) | <0.001 |
| creatinine (mean (SD)) | | 2.08 (1.16) | 0.95 (0.43) | 4.71 (3.07) | 1.20 (0.60) | <0.001 |
| platelet_count (mean (SD)) | | 198.82 (108.42) | 223.04 (101.30) | 184.27 (113.12) | 211.47 (100.92) | <0.001 |
| white_blood_cells (mean (SD)) | | 16.54 (16.49) | 13.30 (6.95) | 15.74 (9.46) | 11.64 (5.45) | <0.001 |
| hemoglobin (mean (SD)) | | 10.74 (2.43) | 12.14 (2.37) | 10.37 (2.49) | 10.52 (2.16) | <0.001 |
| anion_gap (mean (SD)) | | 16.15 (3.99) | 13.79 (3.60) | 22.36 (5.51) | 13.61 (3.26) | <0.001 |
| glucose (mean (SD)) | | 194.17 (124.16) | 160.23 (71.87) | 192.80 (115.67) | 140.75 (48.16) | <0.001 |
| potassium (mean (SD)) | | 4.51 (0.88) | 4.08 (0.66) | 5.02 (1.02) | 4.03 (0.61) | <0.001 |
| sodium (mean (SD)) | | 138.11 (6.24) | 138.16 (5.09) | 136.14 (6.46) | 139.91 (4.92) | <0.001 |
| ph (mean (SD)) | | 7.31 (0.10) | 7.37 (0.08) | 7.24 (0.12) | 7.40 (0.07) | <0.001 |
| po2 (mean (SD)) | | 99.17 (78.95) | 122.32 (91.33) | 105.82 (84.28) | 138.33 (106.31) | <0.001 |
| triglycerides (mean (SD)) | | 202.92 (225.50) | 199.02 (244.55) | 265.31 (343.31) | 132.79 (99.82) | <0.001 |
| phosphate (mean (SD)) | | 4.30 (1.26) | 3.34 (1.00) | 7.01 (2.03) | 3.42 (0.90) | <0.001 |
| los_icu (mean (SD)) | | 12.25 (10.83) | 12.57 (11.99) | 13.15 (12.40) | 9.74 (9.49) | <0.001 |
| eGFR (mean (SD)) | | 41.18 (22.91) | 88.07 (22.62) | 19.79 (15.80) | 62.57 (22.25) | <0.001 |
| scd | 0 | 338 (46.9) | 397 (53.9) | 145 (43.8) | 495 (58.4) | <0.001 |
| scd | 1 | 382 (53.1) | 340 (46.1) | 186 (56.2) | 353 (41.6) | |
| af | 0 | 274 (38.1) | 394 (53.5) | 142 (42.9) | 281 (33.1) | <0.001 |
| af | 1 | 446 (61.9) | 343 (46.5) | 189 (57.1) | 567 (66.9) | |
| pad | 0 | 695 (96.5) | 720 (97.7) | 320 (96.7) | 815 (96.1) | 0.348 |
| pad | 1 | 25 (3.5) | 17 (2.3) | 11 (3.3) | 33 (3.9) | |
| htn | 0 | 494 (68.6) | 414 (56.2) | 255 (77.0) | 480 (56.6) | <0.001 |
| htn | 1 | 226 (31.4) | 323 (43.8) | 76 (23.0) | 368 (43.4) | |
| cva | 0 | 616 (85.6) | 584 (79.2) | 294 (88.8) | 676 (79.7) | <0.001 |
| cva | 1 | 104 (14.4) | 153 (20.8) | 37 (11.2) | 172 (20.3) | |
| ckd | 0 | 456 (63.3) | 695 (94.3) | 214 (64.7) | 612 (72.2) | <0.001 |
| ckd | 1 | 264 (36.7) | 42 (5.7) | 117 (35.3) | 236 (27.8) | |
| t2dm | 0 | 427 (59.3) | 539 (73.1) | 181 (54.7) | 571 (67.3) | <0.001 |
| t2dm | 1 | 293 (40.7) | 198 (26.9) | 150 (45.3) | 277 (32.7) | |
| t1dm | 0 | 703 (97.6) | 729 (98.9) | 322 (97.3) | 841 (99.2) | 0.019 |
| t1dm | 1 | 17 (2.4) | 8 (1.1) | 9 (2.7) | 7 (0.8) | |
| hld | 0 | 434 (60.3) | 466 (63.2) | 213 (64.4) | 432 (50.9) | <0.001 |
| hld | 1 | 286 (39.7) | 271 (36.8) | 118 (35.6) | 416 (49.1) | |
| hf | 0 | 295 (41.0) | 395 (53.6) | 151 (45.6) | 402 (47.4) | <0.001 |
| hf | 1 | 425 (59.0) | 342 (46.4) | 180 (54.4) | 446 (52.6) | |

*(Continued)*

**Table 1.** (Continued)

| Variable | level | 1 | 2 | 3 | 4 | p |
|---|---|---|---|---|---|---|
| n | | 720 | 737 | 331 | 848 | |
| mi | 0 | 530 (73.6) | 569 (77.2) | 228 (68.9) | 722 (85.1) | <0.001 |
| mi | 1 | 190 (26.4) | 168 (22.8) | 103 (31.1) | 126 (14.9) | |
| ihd | 0 | 319 (44.3) | 439 (59.6) | 162 (48.9) | 449 (52.9) | <0.001 |
| ihd | 1 | 401 (55.7) | 298 (40.4) | 169 (51.1) | 399 (47.1) | |
| copd | 0 | 555 (77.1) | 629 (85.3) | 273 (82.5) | 691 (81.5) | 0.001 |
| copd | 1 | 165 (22.9) | 108 (14.7) | 58 (17.5) | 157 (18.5) | |
| used_antidiabetic | 0 | 435 (60.4) | 526 (71.4) | 205 (61.9) | 614 (72.4) | <0.001 |
| used_antidiabetic | 1 | 285 (39.6) | 211 (28.6) | 126 (38.1) | 234 (27.6) | |
| used_lipidlowering | 0 | 649 (90.1) | 660 (89.6) | 303 (91.5) | 730 (86.1) | 0.015 |
| used_lipidlowering | 1 | 71 (9.9) | 77 (10.4) | 28 (8.5) | 118 (13.9) | |
| used_antihypertensive | 0 | 630 (87.5) | 579 (78.6) | 294 (88.8) | 714 (84.2) | <0.001 |
| used_antihypertensive | 1 | 90 (12.5) | 158 (21.4) | 37 (11.2) | 134 (15.8) | |
| death_within_icu_28days | 0 | 440 (61.1) | 583 (79.1) | 164 (49.5) | 610 (71.9) | <0.001 |
| death_within_icu_28days | 1 | 280 (38.9) | 154 (20.9) | 167 (50.5) | 238 (28.1) | |

Note: This table presents the baseline demographic, clinical, and laboratory characteristics of patients stratified into four groups based on serum phosphate trajectory patterns. Continuous variables are presented as means with standard deviations, and categorical variables as counts with percentages. Abbreviations: n, sample size; SD, standard deviation; p, p-value (statistical significance among groups); BMI, body mass index; SOFA, Sequential Organ Failure Assessment; SAPS II, Simplified Acute Physiology Score II; eGFR, estimated glomerular filtration rate; SCD, sickle cell disease; AF, atrial fibrillation; PAD, peripheral artery disease; HTN, hypertension; CVA, cerebrovascular accident (stroke); CKD, chronic kidney disease; T2DM, type 2 diabetes mellitus; T1DM, type 1 diabetes mellitus; HLD, hyperlipidemia; HF, heart failure; MI, myocardial infarction; IHD, ischemic heart disease; COPD, chronic obstructive pulmonary disease.

31.95 ± 9.30). Metabolic parameters further distinguished the clusters. Group 3 had significantly higher levels of first lactate (4.78 ± 4.28 mmol/L), creatinine (4.71 ± 3.07 mg/dL), and anion gap (22.36 ± 5.51 mmol/L) compared to the other groups. In addition, Group 3 displayed notably elevated serum phosphate levels (7.01 ± 2.03 mg/dL) relative to Groups 1 (4.30 ± 1.26 mg/dL), 2 (3.34 ± 1.00 mg/dL), and 4 (3.42 ± 0.90 mg/dL). Measures of kidney function also varied significantly, with Group 3 having the lowest eGFR (19.79 ± 15.80 mL/min/1.73 m²) compared to higher values in the other groups. Notably, Group 3 patients were significantly younger (mean age: 63.53 ± 13.21 years) yet exhibited the highest SOFA (11.35 ± 3.71) and SAPSII (57.59 ± 14.86) scores, along with pronounced metabolic derangements including elevated lactate, creatinine, anion gap, and serum phosphate levels (7.01 ± 2.03 mg/dL), and the lowest eGFR (19.79 ± 15.80 mL/min/1.73 m²). Notably, Group 3 patients were significantly younger (mean age: 63.53 ± 13.21 years) yet exhibited the highest SOFA (11.35 ± 3.71) and SAPSII (57.59 ± 14.86) scores, along with pronounced metabolic derangements including elevated lactate, creatinine, anion gap, and serum phosphate levels (7.01 ± 2.03 mg/dL), and the lowest eGFR (19.79 ± 15.80 mL/min/1.73 m²). Importantly, the 28-day ICU mortality rates varied markedly across the groups, with Group 3 showing the highest mortality of 50.5% (167/331), followed by Group 1 at 38.9% (280/720), Group 4 at 28.1% (238/848), and Group 2 at 20.9% (154/737).

## Serum phosphate trajectory analysis

We subsequently performed a trajectory analysis of serum phosphate levels in High-Risk sepsis–CKM patients. For each patient, we extracted serum phosphate measurements for the 7 consecutive days following ICU admission; when multiple measurements were recorded on the same day, the daily average was used (Table 2). By plotting the changes in serum phosphate trajectories, three distinct patterns were identified. Fig 3 illustrates these patterns: in Trajectory

**Table 2. Baseline characteristics of patients with three distinct serum phosphate trajectory patterns.**

| Variable | Group 1 | Group 2 | Group 3 |
|---|---|---|---|
| age | 68.513 | 70.388528 | 63.455516 |
| sofa | 4.4284 | 9.166667 | 11.427046 |
| sapsii | 35.471 | 53.288961 | 58.24911 |
| lactate | 1.7196 | 2.607467 | 5.35516 |
| creatinine | 1.1069 | 2.012554 | 4.711744 |
| platelet_count | 218.17 | 197.411255 | 189.441281 |
| white_blood_cells | 12.175 | 16.026201 | 16.238078 |
| hemoglobin | 11.296 | 10.732576 | 10.517794 |
| anion_gap | 13.658 | 15.954545 | 22.935943 |
| glucose | 148.01 | 185.211039 | 206.886121 |
| potassium | 4.0257 | 4.473593 | 5.096797 |
| sodium | 139.17 | 138.094156 | 136.010676 |
| ph | 7.3927 | 7.30947 | 7.223096 |
| po2 | 127.1 | 110.821429 | 105.327402 |
| triglycerides | 162.36 | 201.104978 | 267.078292 |
| phosphate | 3.3592 | 4.253355 | 7.270107 |

Notes: Values are presented as means. Group 1 generally reflects a mild phosphate profile, Group 2 reflects moderate dysregulation, and Group 3 reflects severe phosphate elevation. Variables were measured at ICU admission. Abbreviations: SOFA, Sequential Organ Failure Assessment; SAPS II, Simplified Acute Physiology Score II; PaO$_2$, arterial oxygen partial pressure.

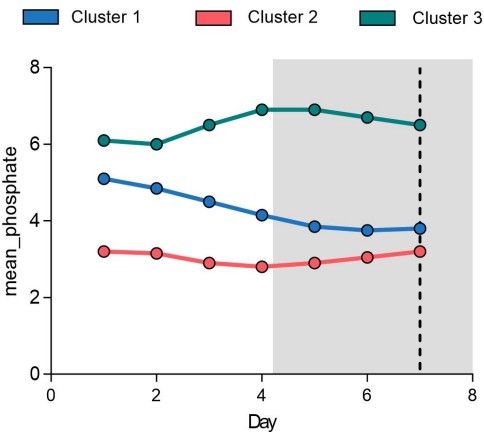

**Fig 3. Phosphate Trajectory-based Patient Clustering.** Patients are classified into three clusters (Cluster 1, 2, and 3) based on their phosphate trajectories during the first 7 days of hospitalization. The line chart shows the mean serum phosphate levels over time, revealing distinct metabolic patterns and potential implications for clinical outcomes.

1, initially high serum phosphate levels gradually declined; Trajectory 2 maintained consistently low serum phosphate levels; and Trajectory 3 showed persistently high serum phosphate levels, with a tendency for further increase. Notably, Trajectory Group 3, characterized by persistently elevated serum phosphate levels, was found to have baseline characteristics nearly identical to those of the young patients with high metabolic derangements identified by consensus clustering (Subtype 3).

## Regression and subgroup analyses

Multivariable logistic regression analysis was performed to assess the association between serum phosphate trajectory patterns and 28-day ICU mortality. Using Group 2 as the reference, patients in Group 3 demonstrated a significantly higher risk of death (OR ≈ 2.9, 95% CI: 2.1–3.0, p<0.001) in the multivariable model. Propensity score-based IPW models yielded consistent results, with Group 3 exhibiting an OR of approximately 4.0 (95% CI: 3.6–4.6, p<0.001) (Table 3). Subgroup analyses were conducted stratifying patients by age (≥65 vs.<65) and gender. In both age and gender subgroups, Group 3 remained significantly associated with higher mortality compared with Group 2. Forest plots (Fig 4) visually illustrate the adjusted odds ratios, 95% confidence intervals, and p-values across these subgroups.

## Discussion

In this study, we identified distinct subtypes based on key clinical and laboratory variables in the MIMIC database. Notably, our analysis showed that Group 3 was characterized by persistently elevated serum phosphate levels despite a lower mean age. Compared with the other subtypes, this group exhibited significantly higher SOFA scores and severer metabolic dysregulation, including elevated lactate, glucose, and triglyceride levels, as well as lower eGFR. Furthermore, compared to the reference group, patients in Group 3 had a significantly higher 28-day ICU mortality risk. These findings suggest that early assessment of serum phosphate trajectories may serve as a valuable surrogate marker for metabolic dysregulation and could assist risk stratification in sepsis.

Sepsis is a complex clinical syndrome, causing muti-organ dysfunction, in which metabolic homeostasis matters as well [19]. Circulation, respiration, kidney and coagulation are usually affected during sepsis. Herein, it is of significance to introduce C-K-M syndrome into sepsis to further Fig out different subtypes. In spite of the development of therapeutic agents and strategies, the mortality rate of sepsis remains high among critically ill patients [20,21]. As a result, several indicators are developed as predictors of sepsis, including old age [22], serum lactate level [23], red blood cell distribution [24], urea nitrogen level [25] and SOFA score. Due to the complexity of sepsis, it is still difficult to significantly improve the prognosis of sepsis. Hence, potential predictors remain further exploration.

Sepsis induced cardiomyopathy is one of the severe complications during sepsis, causing temporary or even permanent cardiac injury and leading to an increase of mortality. Researchers have paid attention to this disease and found overdose inflammation and immune disturbance act as predominant factors in the pathogenesis [26]. In addition, sepsis patients with kidney dysfunction are inclined to present poor prognosis regardless of CKD or AKI. As shown in a large population-based cohort study, CKD was associated with an increased risk of bloodstream infection and related death [27]. Organ dysfunction during sepsis partly attributes to the metabolic alterations, indicating that metabolism affects overall tissues and cells. To supply energy for the inflammatory response rapidly, the body generally shifts its energy metabolism toward glycolysis. While this adaptation is beneficial in the short term for managing inflammation, prolonged reliance on glycolysis often triggers an excessive anti-inflammatory response, ultimately resulting in poor prognosis [28].

**Table 3. Regression Analyses.**

| Method | Cluster | OR | 95% CI | p_value |
|---|---|---|---|---|
| Multivariate | Group 1 | 1.85 | (1.502–2.278) | <0.001 |
| Multivariate | Group 3 | 2.909 | (2.121–2.991) | <0.001 |
| IPW | Group 1 | 1.814 | (1.609–2.046) | <0.001 |
| IPW | Group 3 | 4.026 | (3.557–4.556) | <0.001 |

Notes: Odds ratios (OR) and 95% confidence intervals (CI) were calculated using multivariate logistic regression and inverse probability weighting (IPW). Group 2 was set as the reference category. A p-value <0.05 indicates statistical significance. Abbreviations: OR, odds ratio; CI, confidence interval; IPW, inverse probability weighting.

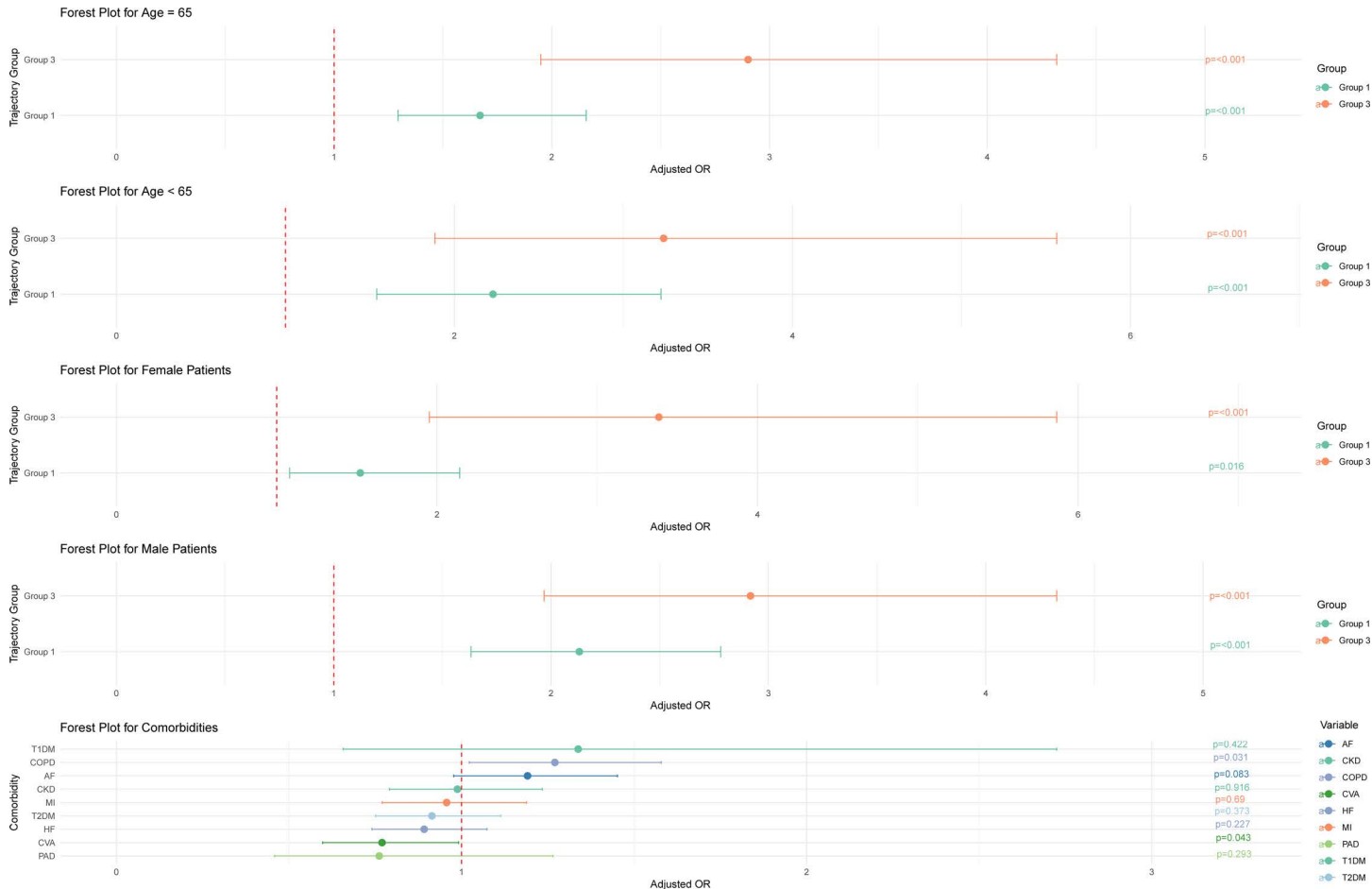

**Fig 4. Patient Groups by Subgroups.** These forest plots display adjusted odds ratios (ORs) and 95% confidence intervals for three patient groups (Group 1, Group 2, Group 3) across various subgroups, including age (>65 vs. ≤65), sex (male vs. female), and comorbidity status. The vertical reference line at OR=1 indicates no difference in risk; estimates to the right suggest higher risk, and those to the left suggest lower risk.

Almost all immune cells undergo metabolic reprogramming during sepsis. Previous studies have also observed metabolic reprogramming in organ cells, such as tubular epithelial cells and cardiomyocytes, wherein glycolysis supplants oxidative phosphorylation [29]. Identification of metabolic disturbances and the regulation of metabolic homeostasis are of paramount importance in improving the prognosis of sepsis patients. Timely intervention to modulate metabolic dysregulation and thereby limit the inflammatory response should be a key focus of future research [30]. As a result, incorporating C-K-M syndrome into sepsis risk stratification would be helpful for sepsis individualize therapy.

Phosphate is engaged in the process of metabolism and also emerges as the product of metabolism. Serum phosphate dynamically changes in the whole body, depending on the state of body. Changes of serum phosphate are suggested to be predictors of adverse outcomes among critically ill patients [5,31,32]. An increase in phosphate level at 48h was indicated to be related to an 8.62-fold increased risk of all-cause mortality in patients with AKI undergoing continuous veno-venous hemodiafiltration [33]. Hyperphosphatemia commonly happens in patients with increased catabolism, tissue destruction, crush injury, rhabdomyolysis, or hyperthermia [34]. Systemic infections caused cellular breakdown and release phosphate from the cells into the extracellular fluid [35]. In heart, increased coronary venous inorganic phosphate concentration was caused by ATP utilization in hypoxic cardiomyocytes. Lactic acid is commonly increased in sepsis, especially septic shock,

making intracellular phosphate transferred into circulation and finally hyperphosphatemia [36]. However, the mechanism that the serum phosphate level and the outcome in sepsis remains to be elucidated. There are several possible explanations. Elevated serum phosphate caused endothelial dysfunction and vascular calcification, contributing to impaired microcirculatory blood flow and organ failure [16,37]. Higher serum phosphate was associated with microvascular dysfunction even in common individuals [38]. Moreover, fibroblast growth factor-23(FGF-23) is an endocrine hormone that regulates phosphate, which increases in parallel with serum phosphate and directly impair leukocyte recruitment and host defense [15]. FGF23 might also affect the immune response directly and indirectly through inflammation. There could be complicated relationship among phosphate, inflammation and immunity. In this study, persistently elevated phosphate levels were significantly associated with increased mortality risk among high-risk CKM-sepsis patients, suggesting phosphate not only serves as a marker of metabolic dysregulation but may actively contribute to CKM disease progression [3]. Moreover, we introduce the emerging metabolic concept of CKM Syndrome. By characterizing CKM-Sepsis comorbid patients, a new perspective on the metabolic management of sepsis is provided. Additionally, we developed a simplified CKM staging system based on BMI, blood glucose, blood pressure, triglyceride levels, and key comorbidities, which further stratified patients into low-, moderate-, and High-Risk categories [39]. Most sepsis patients were classified as Stage 2 to Stage 4b, with the highest risk group (Stage 4a–4b) exhibiting the worst outcomes. Notably, the baseline characteristics of patients in Group 3 from the clustering analysis were closely aligned with those demonstrating marked metabolic dysregulation and severe CKM staging, further reinforcing the potential clinical utility of serum phosphate as an indicator of disease severity.

Firstly, our study utilized a large, well-defined dataset from the MIMIC database, which includes comprehensive clinical and laboratory data. Secondly, the use of consensus clustering allowed for more reliable identification of patient subtypes, overcoming several limitations associated with traditional clustering techniques. Additionally, by analyzing serum phosphate trajectories during the first 7 days of ICU stay, we captured dynamic changes that reflect the continuously evolving metabolic state of patients, potentially providing superior prognostic information. Finally, the development of a simplified CKM staging system, combined with clustering analysis, offers a multifaceted approach to risk stratification that can guide personalized therapeutic interventions.

There are several limitations as well. As a retrospective study, our findings are inherently subject to biases related to data selection and unmeasured confounding factors. While practical, our simplified CKM staging system may not capture all the subtle nuances of metabolic and organ dysfunction in sepsis patients, necessitating prospective validation and refinement. The study only considered serum phosphate trajectories during the first 7 days of ICU admission, potentially overlooking later changes that could affect long-term outcomes.

## Conclusion

In conclusion, our study demonstrates that the integration of a simplified CKM staging system further supported the risk stratification in sepsis and serum phosphate trajectories could serve as a robust marker of metabolic dysregulation in high-risk CKM-Sepsis patients. Future research should focus on prospectively validating the CKM staging system in external ICU cohorts, elucidating the causal role of phosphate metabolism in sepsis progression, and assessing whether targeted modulation of phosphate levels could offer therapeutic benefits in metabolically compromised patients.

## Supporting information

**S1 Fig. S1. K-means Clustering Classification Patients.** This figure presents the results of consensus clustering analysis to determine the optimal number of clusters (k) for Cardiovascular-Kidney-Metabolic-Sepsis patients. (A) Delta area plot: Shows the relative change in the area under the cumulative distribution function (CDF) curve as k increases, indicating the optimal number of clusters. (B) Consensus CDF plot: Displays the cumulative distribution function (CDF) for different k-values, where greater separation and stability of CDF curves suggest the most suitable cluster number. (DOCX)

**S2 Graphic abstract. CKM Syndrome-Sepsis Interplay: Stratification, Targets, and Phosphate Role.** This schematic illustrates the interplay between cardiovascular–kidney–metabolic (CKM) syndrome and sepsis, highlighting patient stratification from lower to higher risk, potential clinical characteristics, and therapeutic targets. It underscores the importance of phosphate metabolism in disease progression and prognosis.
(PNG)

## Acknowledgments

We would like to express our gratitude to all researchers whose work contributed to the foundation of this study. The authors would also like to acknowledge BioRender.com for providing an intuitive platform that facilitated the creation of Fig 1, which illustrates the mitochondrial mechanisms and potential therapeutic targets in sepsis-associated encephalopathy and sepsis-induced cardiomyopathy.

## Author contributions

**Conceptualization:** Zhihong Zuo, Jinwei Dai, Nianzhe Sun, Zhaoxin Qian.

**Data curation:** Zhihong Zuo, Jinwei Dai, Wenye Xu.

**Formal analysis:** Zhihong Zuo, Jinwei Dai.

**Funding acquisition:** Zhihong Zuo, Zhaoxin Qian.

**Investigation:** Zhihong Zuo.

**Methodology:** Zhaoxin Qian.

**Resources:** Jinwei Dai.

**Software:** Jinwei Dai.

**Supervision:** Jinwei Dai, Nianzhe Sun, Ting Wu.

**Validation:** Zhihong Zuo, Jinwei Dai, Nianzhe Sun, Ting Wu.

**Visualization:** Zhihong Zuo, Jinwei Dai, Wenye Xu, Nianzhe Sun, Ting Wu, Zhaoxin Qian.

**Writing – original draft:** Zhihong Zuo, Jinwei Dai, Wenye Xu, Ting Wu, Zhaoxin Qian.

**Writing – review & editing:** Zhihong Zuo, Jinwei Dai, Wenye Xu, Ting Wu, Zhaoxin Qian.

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
