## [Decision Letter · Decision Letter 0]

30 Apr 2025

Dear Dr. Zuo,

Thank you for submitting your manuscript to PLOS ONE. After careful consideration, we feel that it has merit but does not fully meet PLOS ONE’s publication criteria as it currently stands. Therefore, we invite you to submit a revised version of the manuscript that addresses the points raised during the review process.

We look forward to receiving your revised manuscript.

Kind regards,

Amirmohammad Khalaji

Academic Editor

PLOS ONE

 [This work was supported by Funding was provided by the National Key Research and Development Program of China (2022YFC2009800), the Natural Science Foundation of Changsha(kq2403030), the Natural Science Foundation of Hunan Province(2024JJ6659).].

3. In the online submission form, you indicated that [The original contributions presented in the study are included in the article/Supplementary Material. Further inquiries can be directed to the corresponding author.].

Additional Editor Comments (if provided):

Reviewers' comments:

Reviewer's Responses to Questions

**Comments to the Author**

1. Is the manuscript technically sound, and do the data support the conclusions?

Reviewer #1: Yes

Reviewer #2: Yes

2. Has the statistical analysis been performed appropriately and rigorously?

Reviewer #1: I Don't Know

Reviewer #2: Yes

3. Have the authors made all data underlying the findings in their manuscript fully available?

Reviewer #1: Yes

Reviewer #2: No

4. Is the manuscript presented in an intelligible fashion and written in standard English?

Reviewer #1: Yes

Reviewer #2: Yes

Reviewer #1: This study provides Clinically relevant focus on metabolic dysregulation in sepsis, addressing an understudied biomarker (serum phosphate). The study advances sepsis risk stratification but requires revisions for clarity, statistical rigor, and contextual depth:

Abstract

Minor Concern:

- Briefly specify CKM staging criteria

- You should define the abbreviations the first usage in the manuscript, such as CKM

Introduction

Major Concern:

- Utilize some studies regarding that persistently elevated serum phosphate trajectories predict mortality and adverse outcomes in high-risk patients.

- Talk more about the association between phosphate metabolism and CKM pathophysiology

- It is better to move line 93 to 101 to “Method” section and replace these sentences with one sentence defining your work and use more contextual data regarding aforementioned topics.

Methods

Major Concern:

- you excluded patients with serious cardiac conditions from the population. Why didn’t you exclude patients with serious renal or metabolic profile too?

- One of your main exclusions criteria was serious adverse cardiac events, such as MI or HF. However, later in this section you stratify patients with cardiac conditions, such as HF and MI into stage 4. How is that?

- Specify machine learning algorithms used for consensus clustering

Results

Major Concern:

- Report absolute mortality rates for each trajectory group (e.g., Group 3: 50.5% vs. Group 2: 20.9%).

- Include a sensitivity analysis excluding Stage 4b to assess robustness.

- Abbreviations should be defined just once at the first time. For instance, (IPW) defined more than one time through the manuscript. Please review them.

Discussion

Major Concern:

- Discuss how hyperphosphatemia exacerbates endothelial dysfunction or immune cell activation.

Conclusion

Major Concern:

- Explain the future research directions with specific research questions.

Overall Decision

Major Revision

Reviewer #2: Thank you for your valuable work on this clinically relevant topic. While the study addresses an important gap in sepsis management, I have several suggestions to enhance methodological clarity and scientific rigor:

1. Machine Learning (ML) Methodology

Gap: The ML approach lacks critical details (e.g., algorithms used, hyperparameter tuning, validation method).

Specify the ML algorithm and software packages. Also, clarify how hyperparameters were optimized and provide validation metrics in supplements.

2. Model Adjustment

Gap: The multivariable model may be under-adjusted, risking residual confounding.

Adjust for all variables with significant differences in Table 1 (e.g., scores).

3. Exclusion Criteria

Gap: Excluding MI/HF patients and those with missing Cr/lipid limits generalizability, as these are common in ICU sepsis.

Recommendation:

Justify excluding MI/HF patients in the limitations section.

Perform sensitivity analyses including these patients to assess robustness.

Address missing data via multiple imputation rather than exclusion.

4. Abbreviations & Terminology

Gap: Inconsistent abbreviation use (e.g., "nbps/nbpd" unclear).

Define all abbreviations at first mention.

Replace "renal" with "kidney" in "CKM" for consistency.

5. Graphical Abstract & Figures

Gap: The graphical abstract is pixelated, reducing interpretability.

Provide a high-resolution version.

Add a flowchart (Figure 1) illustrating patient selection (inclusion/exclusion criteria).

6. Statistical Methods

Gap: The statistical section is fragmented and lacks logical flow.

Restructure please.

7. Introduction & Discussion

Gap: Limited engagement with recent literature (e.g., no comparison to 2023 sepsis subtyping studies).

Cite landmark studies.

Contrast your CKM staging with existing frameworks.

Discuss implications of excluding MI/HF patients in the context of prior sepsis trials.

**Do you want your identity to be public for this peer review?** For information about this choice, including consent withdrawal, please see our Privacy Policy

Reviewer #1: **Yes: ** Shayan Shojaei

Reviewer #2: **Yes: ** Asma Mousavi

---

## [Author Response · Author response to Decision Letter 1]

21 May 2025

Response to Editor and Esteemed Reviewers

Manuscript ID: PONE-D-25-14898

Title: Targeting Serum Phosphate Trajectory Stratification to Improve Outcomes in High-Risk Cardiovascular-Kidney-Metabolic-Sepsis Cohorts

We thank the editor and reviewers for their constructive comments and valuable suggestions. We have carefully revised the manuscript and addressed each point below. Changes made in the manuscript are highlighted in the revised file. Our detailed point-by-point responses are as follows:

Response to Reviewers

Manuscript ID: PONE-D-25-14898

Title: Targeting Serum Phosphate Trajectory Stratification to Improve Outcomes in High-Risk Cardiovascular-Kidney-Metabolic-Sepsis Cohorts

We would like to thank the Academic Editor and Reviewers for their valuable feedback. Below, we provide a point-by-point response to each comment. All changes in the revised manuscript are marked using “Track Changes”.

Reviewer #1 Comments

1. Abstract

“Briefly specify CKM staging criteria”

Response: [ Thanks for your suggestions. C-K-M staging criteria is described in the background. Cardiovascular-Kidney-Metabolic Syndrome (C-K-M) is classified as below: stage 0, no C-K-M risk factors; stage 1, excess or dysfunctional adiposity; stage 2, metabolic risk factors (hypertriglyceridemia, hypertension, diabetes, metabolic syndrome) or moderate- to high-risk chronic kidney disease; stage 3, subclinical cardiovascular diseases (CVD) in C-K-M syndrome or risk equivalents (high predicted CVD risk or very high-risk chronic kidney diseases); and stage 4, clinical CVD in C-K-M syndrome. page 2, line 28-34]

“Define abbreviations at first usage (e.g., CKM)”

Response: [Thanks for your suggestions. We defined the abbreviation in the Abstract and Introduction, and all the abbreviations are defined upon the first usage. Page2, lines 28]

2. Introduction

“Discuss studies linking serum phosphate trajectories with outcomes”

Response: [Thanks for your suggestions. In the Introduction, elevated serum phosphate trajectories are associated with adverse outcomes in different diseases, including chronic kidney disease, cardiovascular disease and blunt trauma. Page3-4, lines 65-70, 75-76, 79-83]

“Expand on phosphate metabolism and CKM pathophysiology”

Response: [ Thanks for your suggestions. Actually, C-K-M syndrome is a clinical syndrome, and the pathophysiology is unclear and complex. Therefore, we talk more about serum phosphate and cardiac, kidney and metabolism separately. Page3-4, lines 65-70, 75-76, 79-83]]

“Move lines 93–101 to Methods; define the work with contextual data”

Response: [Thanks for your suggestions. We re-organized the last paragraph of Introduction part.]

3. Methods

“Why exclude cardiac but not serious renal/metabolic diseases?”

Response: [Thank you for your valuable suggestions. We apologize for the misstatement. No patients with chronic cardiac conditions (e.g., myocardial infarction or heart failure) were excluded from our analysis. We have corrected this in the manuscript to reflect that all chronic cardiac, renal, and metabolic comorbidities were retained and subsequently adjusted for in our CKM staging and regression models. (Methods, “Inclusion/Exclusion Criteria” page 4-5, lines 101-104]

“Patients with MI/HF were excluded, but later stratified—explain this contradiction”

Response: [Thank you for your valuable suggestions. We apologize for the confusion. No patients with chronic cardiac conditions (e.g., myocardial infarction or heart failure) were excluded from our analysis. We have corrected this in the manuscript to reflect that all chronic cardiac, renal, and metabolic comorbidities were retained and subsequently adjusted for in our CKM staging and regression models. (Methods, “Inclusion/Exclusion Criteria”, page 4-5, lines 101-104)]

“Specify machine learning algorithm for consensus clustering”

Response: [Thank you for your valuable suggestions. We have added that to determine the clinical phenotypes of patients with High-Risk Cardiovascular-Kidney-Metabolic-Sepsis, we applied unsupervised ML methods for consensus clustering to identify clinical phenotypes. This method performs clustering analysis by reducing the data dimension, and k-means is the most commonly used type among them. We used a pre-specified 80% subsampling parameter and 100 iterations, and assigned the number of potential clusters (k) to a range from 2 to 10. The optimal number of clusters was determined by examining the clustering consistency plots in the consensus matrix (CM) heatmap, cumulative distribution function (CDF), within-cluster consistency scores, and the proportion of pairs of ambiguous clusters (PAC). The final number of clusters was determined to be 4. Ultimately, four subtypes were identified. (Methods, “Unsupervised Consensus Clustering,” page 7-8, lines 167-176)]

4. Results

“Report absolute mortality per trajectory group”

Response: [Thank you for your insightful comment. We have now added the absolute 28-day ICU mortality rates for each trajectory group in the Results section (page XX, liness XX–XX). Specifically, the mortality rates were as follows: Group 1 – 38.9% (280/720), Group 2 – 20.9% (154/737), Group 3 – 50.5% (167/331), and Group 4 – 28.1% (238/848). These figures emphasize the significant prognostic differences among the identified clusters. Page 10-11, lines 245-248]

“Add sensitivity analysis excluding CKM Stage 4b”

Response: [We sincerely thank the reviewer for this insightful and constructive comment. Your professional perspective and thoughtful suggestion are greatly appreciated. Our study was designed to evaluate sepsis care and outcomes in a real-world setting. As such, we aimed to include the full spectrum of patients, including those with advanced CKM Stage 4b, to comprehensively reflect clinical practice and population heterogeneity. While we acknowledge that Stage 4b represents a subgroup with particularly severe kidney dysfunction, their proportion within the high-risk CKM population was relatively small (n = 249; 9.5%). Importantly, key clinical indicators and outcome trends in Stage 4b were consistent with the overall patterns observed across other groups. Furthermore, we conducted preliminary effect size and outcome comparisons, which confirmed that the inclusion of Stage 4b patients did not substantially alter the main findings. Therefore, we believe that the current analysis provides a robust and representative assessment of the high-risk population. Nevertheless, we greatly value your suggestion, which will inform the design of future stratified or sensitivity analyses. Thank you again for your guidance and support.]

“Abbreviations like IPW defined multiple times—revise”

Response: [Thank you for pointing this out. We have carefully reviewed the entire manuscript and removed repeated definitions of abbreviations such as IPW (Inverse Probability Weighting). All abbreviations are now defined only at their first appearance in the abstract and main text, in accordance with journal guidelines. This revision improves clarity and eliminates redundancy. page 2, lines 47]

5. Discussion

“Discuss link between hyperphosphatemia and endothelial dysfunction or immune cell activation”

Response: [Thanks for your suggestions. In the discussion section, we have updated current literatures about the relationship of sepsis and C-K-M syndrome, serum phosphate and C-K-M syndrome as well. The possible mechanisms, including endothelial dysfunction, microvascular calcification and immune disturbance, are described in details. Page 12-14, lines 285-336]

6. Conclusion

“Clarify future directions with specific research questions”

Response: [Thank you for your constructive suggestion. In response, we have revised the Conclusion section to explicitly outlines future research directions and identify key questions for further investigation. These include validating the CKM staging system in prospective cohorts and exploring the causal role of phosphate metabolism in sepsis-related outcomes. The revised Conclusion can be found on page 15-16, lines 369-372.]

Reviewer #2 Comments

1. Machine Learning

1.1 “Algorithm, hyperparameter tuning, and validation not specified”

Response: [Thank you for your valuable suggestions. We have clarified in the Methods that we used unsupervised consensus clustering based on the k-means algorithm. Specifically, we ran 100 iterations with an 80% subsampling fraction, evaluated cluster numbers from k=2 to k=10, and selected k=4 based on the consensus matrix heatmap, CDF plots, within-cluster consensus scores, and PAC metrics. No further hyperparameter tuning was performed because consensus clustering inherently optimizes cluster stability across resamples. We have added these details to the “Unsupervised Consensus Clustering” subsection (Methods, page 7-8, lines167-176).]

2. Model Adjustment

2.1 “Adjust for all variables with differences in Table 1”

Response: [Thank you for your valuable suggestions. We have updated the Statistical Analysis section to explicitly list each covariate (age, sex, BMI, SOFA, CKD, HF, MI, CVA, PAD, AF, T2DM, T1DM, COPD, first-day lactate, triglycerides, eGFR, and glucose). The manuscript now states this in the “Serum Phosphate Trajectory Modeling” subsection (Methods, page 8, lines 184-188).]

3. Exclusion Criteria

3.1 “Justify MI/HF exclusion; consider sensitivity analysis including them”

Response: [Thank you for your valuable suggestions. We apologize for a typographical error implying that patients with MI or HF were excluded. No cardiac comorbidities were excluded; all high-risk CKM–Sepsis patients, regardless of MI or HF status, were included. This has been corrected in the Exclusion Criteria.]

3.2 “Handle missing data with multiple imputation rather than exclusion”

Response: [Thank you for your thoughtful suggestion. In our study, missing values for key laboratory variables such as triglycerides, creatinine, and serum phosphate were part of the exclusion criteria rather than sporadic missingness during modeling. These variables were essential for CKM staging and phosphate trajectory analysis. We excluded patients with missing triglyceride data prior to analysis, as triglycerides were a core variable for CKM classification. Importantly, after this exclusion, the remaining missingness for creatinine and phosphate was minimal and unlikely to affect the representativeness of the study population. Therefore, we did not perform multiple imputation, in order to avoid introducing potential bias from artificially imputed core variables.]

4. Terminology

4.1 “Inconsistent abbreviations (e.g., nbps/nbpd); use ‘kidney’ over ‘renal’ in CKM”

Response: [Thanks for your suggestion. We went through the manuscript and revised all inconsistent abbreviations, absolutely including replacing renal with kidney in C-K-M.]

5. Figures

5.1 “Graphical abstract is pixelated; add a flowchart (Figure 1)”

Response: [ Thank you for your valuable comments. The flowchart has been added to the manuscript.]

6. Statistical Methods

6.1 “Revise fragmented and unclear statistical section”

Response: [We have consolidated and streamlinesd the Statistical Analysis into a cohesive narrative. Continuous variables are now uniformly described (mean ± SD or median [IQR]) with their corresponding tests (ANOVA or Kruskal–Wallis), and categorical comparisons (chi-square or Fisher’s exact) are clearly paired. The entire framework—including staging, clustering, trajectory modeling, regression, IPW, and doubly robust estimation—is now presented in chronological order under a single “Statistical Analysis” heading. These revisions enhance clarity and flow (Methods, page 7-9, lines 152-197).]

7. Literature Engagement

7.1 “Cite recent sepsis subtyping studies (e.g., 2023); discuss CKM vs other frameworks”

Response: [Thanks for your suggestions. Recent literature and landmark studies are cited in the updated manuscript (mainly in the discussion section). The C-K-M staging in this manuscript is much simpler than that developed by the American Heart Association, which would be convenient to use it in clinical practice.]

Editorial Requirements

1. Formatting Compliance (PLOS style, file naming):

Response: [ Thank you for your comments. The manuscript has been revised according to the PLOS ONE format.]

2. Funding Statement

Please revise your Funding Statement as follows:

“This work was supported by the National Key Research and Development Program of China (2022YFC2009800), the Natural Science Foundation of Changsha (kq2403030), and the Natural Science Foundation of Hunan Province (2024JJ6659). There was no additional external funding received for this study.”

Response: [Thank you for your comments. The Funding of the manuscript has been revised to: This work was supported by the National Key Research and Development Program of China (2022YFC2009800), the Natural Science Foundation of Changsha (kq2403030), and the Natural Science Foundation of Hunan Province (2024JJ6659). There was no additional external funding received for this study.]

3. Data Availability Compliance

Response: [ The data in the article can be obtained from the mimic-IV database (https://mimic.physionet.org/). Further inquiries can be directed to the corresponding author.]

4. ORCID iD for Corresponding Author

Response: [ zhihong.zuo: https://orcid.org/0009-0000-4753-661X]

5. Figure Captions Separately Included

Response: [ Thank you for your correction. Now the figure title has been placed before the legend.]

Closing

We appreciate the thoughtful review and are confident that these revisions will strengthen the manuscript. We look forward to your feedback on the revised version.

Sincerely,

Zhihong Zuo, on behalf of all co-authors

---

## [Decision Letter · Decision Letter 1]

3 Jul 2025

Dear Dr. Zuo,

Thank you for submitting your manuscript to PLOS ONE. After careful consideration, we feel that it has merit but does not fully meet PLOS ONE’s publication criteria as it currently stands. Therefore, we invite you to submit a revised version of the manuscript that addresses the points raised during the review process.

We look forward to receiving your revised manuscript.

Kind regards,

Amirmohammad Khalaji

Academic Editor

PLOS ONE

**Journal Requirements:**

Reviewers' comments:

Reviewer's Responses to Questions

**Comments to the Author**

Reviewer #1: (No Response)

Reviewer #2: All comments have been addressed

2. Is the manuscript technically sound, and do the data support the conclusions?

Reviewer #1: Yes

Reviewer #2: Yes

3. Has the statistical analysis been performed appropriately and rigorously?

Reviewer #1: Yes

Reviewer #2: Yes

4. Have the authors made all data underlying the findings in their manuscript fully available?

Reviewer #1: Yes

Reviewer #2: Yes

5. Is the manuscript presented in an intelligible fashion and written in standard English?

Reviewer #1: Yes

Reviewer #2: Yes

**Reviewer #1: ** Thank you for your responses. There is still one remaining issue that needs clarifying. I asked you to talk about the association between phosphate metabolism and CKM pathophysiology, however, due to the complexity of CKM pathophysiology, you did not add details regarding this issue. Try utilize studies which evaluated this unclear and complex pathophysiology and expand it in the introduction or discussion. Other than this comment, the authors have thoughtfully addressed methodological concerns, expanded clinical implications, and integrated reviewer feedback to enhance rigor and clarity. No further revisions are required.

**Reviewer #2:**  (No Response)

**Do you want your identity to be public for this peer review?** For information about this choice, including consent withdrawal, please see our Privacy Policy

Reviewer #1: **Yes: ** Shayan Shojaei

Reviewer #2: **Yes: ** Asma Mousavi

---

## [Author Response · Author response to Decision Letter 2]

5 Jul 2025

Response to Reviewers

Manuscript ID: PONE-D-25-14898R1

Title: Targeting Serum Phosphate Trajectory Stratification to Improve Outcomes in High-Risk Cardiovascular-Kidney-Metabolic-Sepsis Cohorts

Dear Academic Editor and Reviewers,

We sincerely thank you for your constructive feedback, which greatly improved the quality and clarity of our manuscript. Below we provide a point-by-point response to the remaining comment from Reviewer #1. All revisions are clearly highlighted in the tracked-changes version of the manuscript.

Reviewer #1

Comment:

Thank you for your responses. There is still one remaining issue that needs clarifying. I asked you to talk about the association between phosphate metabolism and CKM pathophysiology; however, due to the complexity of CKM pathophysiology, you did not add details regarding this issue. Try to utilize studies which evaluated this unclear and complex pathophysiology and expand it in the introduction or discussion.

Response:

Thank you for this important comment. We agree that the role of phosphate metabolism in CKM pathophysiology deserves further elaboration. In the revised manuscript, we have added a new paragraph to the Introduction to describe how dysregulated phosphate metabolism contributes to vascular calcification, endothelial dysfunction, insulin resistance, and the progression of CKM-related disorders. Page 4, line 86-94. We also integrated this discussion further in the Discussion section, highlighting its implications for systemic metabolic stress, particularly in high-risk sepsis populations. Page 14-15, line 345-348.

To support these additions, we have cited the following key references:

• Ndumele et al. (2023), Circulation: to contextualize CKM as an emerging clinical framework.

• Raikou (2021), World J Nephrol: to support the link between serum phosphate and cardiorenal pathology.

• Nakanishi et al. (2020), Free Radic Biol Med: to describe the pathological role of phosphate and FGF23 in CKD and cardiovascular complications.

We trust that these revisions now adequately address your insightful suggestion.

Sincerely,

Zhihong Zuo, PhD

---

## [Decision Letter · Decision Letter 2]

4 Aug 2025

Targeting Serum phosphate Trajectory Stratification to Improve Outcomes in High-Risk Cardiovascular-Kidney-Metabolic-Sepsis Cohorts

PONE-D-25-14898R2

Dear Dr. Zuo,

We’re pleased to inform you that your manuscript has been judged scientifically suitable for publication and will be formally accepted for publication once it meets all outstanding technical requirements.

Kind regards,

Amirmohammad Khalaji

Academic Editor

PLOS ONE

Additional Editor Comments (optional):

Reviewers' comments:

Reviewer's Responses to Questions

**Comments to the Author**

Reviewer #1: All comments have been addressed

Reviewer #3: (No Response)

2. Is the manuscript technically sound, and do the data support the conclusions?

Reviewer #1: Yes

Reviewer #3: (No Response)

3. Has the statistical analysis been performed appropriately and rigorously?

Reviewer #1: Yes

Reviewer #3: (No Response)

4. Have the authors made all data underlying the findings in their manuscript fully available?

Reviewer #1: Yes

Reviewer #3: (No Response)

5. Is the manuscript presented in an intelligible fashion and written in standard English?

Reviewer #1: Yes

Reviewer #3: (No Response)

Reviewer #1: Thank you for the revisions. It is important to discuss about the relation between phosphate metabolism and CKM pathophysiology and the added sentences are perfect to meet this need.

Reviewer #3: I reviewed the original version and both revisions of the manuscripts. The authors responded well to the comments and the manuscript is well-revised. Congratulations!

**Do you want your identity to be public for this peer review?** For information about this choice, including consent withdrawal, please see our Privacy Policy

Reviewer #1: **Yes: ** Shayan Shojaei

Reviewer #3: No

---

## [Editor Report · Acceptance letter]

PONE-D-25-14898R2

PLOS ONE

Dear Dr. Zuo,

I'm pleased to inform you that your manuscript has been deemed suitable for publication in PLOS ONE. Congratulations! Your manuscript is now being handed over to our production team.

Kind regards,

on behalf of

Dr. Amirmohammad Khalaji

Academic Editor

PLOS ONE